# A Review on Medical Waste Management: Treatment, Recycling, and Disposal Options

**Mustafa Attrah *** , **Amira Elmanadely** , **Dilruba Akter** and **Eldon R. Rene ***

Department of Water Supply, Sanitation and Environmental Engineering, IHE Delft Institute for Water Education, Westvest 7, 2611 AX Delft, The Netherlands
* Correspondence: mustafa.attrah@gmail.com (M.A.); e.raj@un-ihe.org (E.R.R.)

**Abstract:** Many nations struggle with the collection, separation, and disposal of medical waste. However, extra caution is required to avoid the risk of injury, cross-contamination, and infection; thus, healthcare workers and individuals responsible for waste management must follow the mandatory safety procedures. In this review, a classification of the various types and categories of medical waste and its treatment methods are discussed. Due to the fact that medical waste can be contaminated and hazardous, it must be managed and processed using complex steps and procedures. In many countries, the primary medical/hospital waste treatment method is incineration, which is regarded as a highly polluting process that emits numerous pollutants that degrade air quality and pose a threat to human health and the environment. As case studies, medical waste treatment and disposal practices in Germany, China, USA, and Egypt were compared, and the legislations and laws enacted to regulate medical waste in each of these countries are reviewed and discussed.

**Keywords:** medical waste; human health; legislations; classification of medical waste; incineration

## 1. Introduction

The term "medical waste" is used in many countries, such as the US, South Korea, and China, while the European Union and World Health Organisation (WHO) refer to it as "healthcare waste" [1]. The World Health Organization (WHO) defines healthcare waste (medical waste) as any waste or by-products from hospitals and health care facilities for humans and animals used for diagnosis, treatment, or immunisation, e.g., used syringes, needles, metal sharps, dressings, blood samples, body parts, pharmaceutical, chemical, radioactive materials, and devices [2]. Generally, countries with high revenue generate up to 0.5 kg/hospital bed of hazardous medical waste [3]. The health care sector's waste extensively impacts the environment and public health, proving very costly. In addition, the manufacturing and discarding of medical and health care sector waste lead to increased levels of GHG emissions and pollution [4]. The types of plastics that are mainly used to make operating room tools and equipment are polyvinylchloride (PVC), polyethylene (PE), polypropylene (PP), polyurethane (PU), and copolymers. The first three types of plastics can and are being recycled. In general, most of the operating room's waste can be considered non-hazardous because it is generated even before the patient arrives and is not contaminated or infected [5].

Face masks make up a considerable part of the medical waste (MW), especially after the massive increase in use due to COVID-19 and mandatory face mask-wearing regulations. According to the findings of a recent study that included seven hospitals and medical centres in the state of Massachusetts, USA, along with three veterinary hospitals [6], plastic waste accounted for ~30% of the total wastes produced by hospitals. Non-woven polyurethane, polypropylene, or polyacrylonitrile fabrics are used to produce face masks. However, these aforementioned materials are not readily degradable but decompose into smaller pieces and particles into microplastics. Additionally, the use of hand gloves made

of latex or plastic for protection by ordinary people and workers in various sectors after the pandemic led to an increase in the amount of disposed of gloves. In addition, gloves also contribute to pollution of the environment when disposed of improperly because they are made of unrecyclable and undegradable materials [7,8]. A study highlighted that 15% of the total global carbon budget is attributed to the greenhouse gas (GHG) emissions resulting from the life cycle of plastics [9]. Therefore, poor management and disposal of plastics threaten the ability of the global community to meet carbon emissions targets and combat climate change [10].

Sharma et al. [11] reviewed the detrimental impacts of incineration of MW caused by the ashes and gaseous emissions. A vast variety of pollutants are released from a MW incinerator, including fly ashes as particulate matter (PM), carbon monoxide (CO), heavy metals, e.g., arsenic, chromium, nickel, cadmium, copper, lead, etc., acid gases such as sulphur dioxides, nitrogen oxides, and hydrogen chloride, and organic compounds such as carbon tetrachloride ($CCl_4$), benzene, toluene, xylenes, and polycyclic aromatic hydrocarbons. In addition, leachable organic compounds form bottom residues and ashes containing heavy metals and dioxins. In addition, there is the carbon footprint of transportation, autoclave decontamination, thermal treatment (i.e., low and high-temperature incineration at $\geq$850 °C and 1000 °C, respectively), plus the carbon emissions produced during recycling [12]. A study in the UK has found considerable variation between different methods of MW treatment. It was estimated that the carbon footprint of MW treatment by high-temperature incineration was 1074 kg $CO_2$e/t, making the choice of waste treatment method have an impact on the carbon footprint of up to 50-fold [12].

A research study that included three hospitals in the Netherlands found out that hospitals managed to earn more than 39,000 € in only six months from recycling and reusing refurbished tools and materials collected from these hospitals. This profit can encourage the hospitals and the healthcare sector to implement recycling practices and make the waste treatment processes more circular than linear [13]. The specific objectives of this review are as follows: (i) to classify and categorise the different types of medical wastes generated from healthcare facilities, (ii) to outline the steps and processes involved in the disposal, segregation, and treatment of MW, and (iii) to compare the practices for management and treatment of MW in four countries from different parts of the world and define the least deleterious methods and, thus, help decision-makers in the health sector and industry to make better choices, and, finally, to demonstrate the impact of the COVID-19 outbreak on the amounts of waste and the consequences behind it.

## 2. Classification of Medical Waste

According to estimates by the WHO, 15 to 20% of medical wastes can be classified as hazardous materials due to their infectivity, toxicity, and, sometimes, radioactivity [14,15]. However, medical waste management practices are not constant or standardised in all countries because this categorisation is not very clear or decisive [16].

Medical waste refers directly or indirectly to infectious, toxic, or otherwise hazardous waste (HMW), illustrated in Figure 1 and described with examples in Table 1. Medical institutions generate this type of waste during medical or preventative care and related activities, specifically infectious, pathological, damaging, pharmaceutical, and chemical waste [1]. On the other hand, non-hazardous medical waste (NHMW) includes all different regular non-infectious fractions of waste, such as municipal solid waste. HMW is usually contaminated with pathogens. Therefore, it can cause a wide range of infections and diseases in the case of misuse or poor handling and discarding. Adding to that, it can cause environmental contamination in the case of poor management, causing pollution to land, water, plants, animals, and air, leading to the spread of diseases.

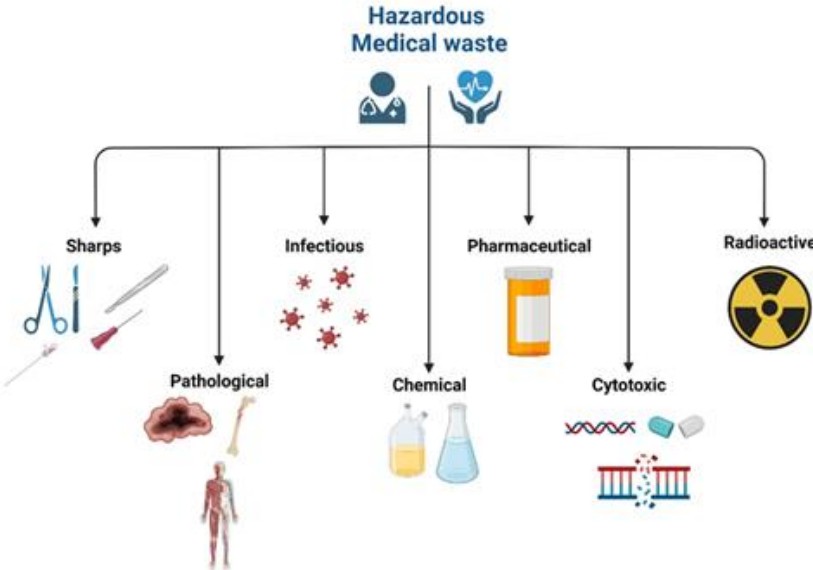

**Figure 1.** Different types of hazardous medical waste.

**Table 1.** Healthcare waste categorisation according to WHO and the EU [3,17].

|  | Category (Examples) | World Health Organization (WHO) | EU | Source |
|---|---|---|---|---|
| Hazardous | Sharps | Sharps | Sharps | Hospitals, clinics, laboratories, blood banks, nursing homes, veterinary clinics and labs |
|  | Organic matter, including body parts and blood | Pathological | Human tissue, body parts, organs, and blood preserves and bags | Hospitals, clinics, laboratories, mortuary and autopsy facilities, veterinary clinics and labs |
|  | Waste with restrictions in collection and disposal due to infectivity | Infectious | Human and Animal Infectious | Hospitals, clinics, and laboratories |
|  | Waste with no restrictions or special requirements for collection and disposal due to infectivity (e.g., plasters, casts, dressings, bed sheets, disposable clothing, etc.) | Infectious | Infectious | Hospitals, clinics, and laboratories |
|  | Dangerous chemical materials and substances | Chemical | Chemical | Hospitals, clinics, and laboratories |
|  | Other chemicals | Chemical | Chemical/ Unused hazardous medicines | Hospitals, clinics, and laboratories |
| Non-hazardous | Cytotoxic and cytostatic medicines | Cytotoxic | Discarded unused medicines | Hospitals and laboratories |
|  | Other chemicals (non-hazardous) | Pharmaceutical | Unused non-hazardous medicines | Hospitals, clinics, and laboratories |
|  | Dental clinics (care centres) amalgam waste | Amalgam (tooth filling) waste from dental clinics/centres | Amalgam waste from dental clinics/centres | Dental care centres and clinics |

MW can also affect physical and mental health and patients' and health workers' quality of life [18]. The plastic portion of MW is approximately 20% to 30% [19].

## 3. Medical Waste Management Process

Medical waste management is a series of steps where the MW generated is handled from the generation point until it can be disposed of safely. The steps of the MW management process are shown in Figure 2. The success of the waste management process is demonstrated in limiting the waste going for disposal and achieving a circular economy, where the materials used within the medical system are maximally utilised, reaching almost zero waste.

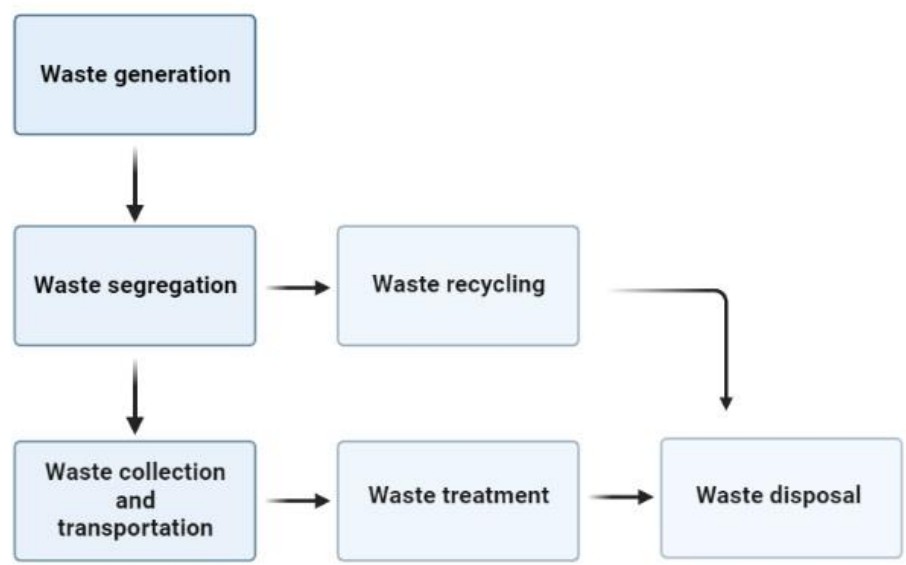

**Figure 2.** Flow scheme for medical waste management.

### 3.1. Waste Generation

The medical waste generated is of various classifications, as mentioned in the previous section. The critical aspect of this step is the amount of waste produced and how it is handled to prevent hazards to the personnel in contact with it. The waste generated from medical institutions can be minimised to reduce waste accumulation. The minimisation can be approached from different directions, such as reduction in waste at source, recycling, and stock management. The reduction in waste at the source can be achieved by reusing materials that will not harm the users, such as washable tablecloths, tableware, and refill containers for cleaning supplies [20]. Recycling can aid in minimising waste by recycling plastics and metals and composting food waste. Finally, stock management will help set an organised system of the medicines inventory to prevent duplication and purchase of unnecessary products that could expire, thus reducing potential waste [20].

### 3.2. Waste Segregation

Segregation identifies the various types of waste and how they can be collected separately. Segregation is mainly achieved by separating different categories of MW in different colour bins or bags specified for each category [20]. Therefore, this causes a problem when collecting from various sources due to the lack of standardisation of the colours associated with each waste category [15], thus increasing the needed time and financial cost of labour and equipment to separate the waste and direct it to the proper waste stream, resulting in a problem. The segregation should mostly occur at the point source except for the waste undergoing the same treatment, which could be separated in the treatment facility [20]. The sharp objects should always be separated at the source [20]. The segregation is carried out by medical staff, which requires training to safely dispose of waste to avoid infections [15]. If a mistake occurs while segregating waste, it should not be corrected to prevent the contamination of the other waste [20]. Medical waste should be stored safely to avoid unauthorised human contact, which can cause infections [15].

### 3.3. Waste Collection and Transportation

The frequency of MW collection should be as high as once per day to avoid the accumulation of waste, which can spread infections. In addition, the personnel responsible for collection should be equipped with safety gear to prevent contaminations and infections that should be safely disposed of after [20]. The waste is collected from the health care entity and transported using secondary transportation to the treatment facility for disposal, recycling, and treatment processes. Treatment facilities are either located within the health care facility or off-site in a separate location [15].

### 3.4. Waste Treatment

Medical waste treatment is a process carried out before the disposal of MW to limit the hazardous effects of this type of waste on the environment and health. The lack of proper treatment can have several impacts, as follows [21]:

- Poisoning from toxic elements,
- Bacterial and fungal infections,
- Release of toxins into the atmosphere,
- Leaching to the soil and underlying aquifers,
- Bioaccumulation,
- Leaving a footprint on the environment,
- Destruction of habitats.

In the production phase of any medical equipment, the impact of these types of equipment must be considered by performing a life cycle analysis (LCA) and practicing proper treatment techniques. However, the methods and techniques for treatment have minimal impact in terms of carbon emissions released into the ambient air. For example, a single intravitreal injection causes the release of 0.05 kg $CO_2$e during the disposal phase [22].

Several treatment methods for MW are illustrated in Table 2 with their primary advantages and disadvantages. Despite these challenges, these methods can reduce the hazards mentioned earlier. The treatments currently in the field are incineration, autoclave disinfection, microwave disinfection, and mechanical and chemical disinfection [21].

**Table 2.** A comparison between treatment methods in terms of advantages and disadvantages.

| Treatment Method | Advantages | Disadvantages |
| --- | --- | --- |
| Incineration | − Weight and volume reduction<br>− Suitable for all waste types<br>− Heat recovery | − Emissions, e.g., furans and dioxins<br>− Public opposition<br>− High capital and operating costs<br>− High maintenance costs<br>− Restrictions due to emissions regulations |
| Autoclave disinfection | − Low operation costs<br>− Adequacy for biological testing<br>− Less hazardous residues | − No change in waste characteristics<br>− Inapplicable for all waste types<br>− Unknown air emissions |
| Microwave disinfection | − Volume reduction<br>− No liquid discharge | − High capital cost<br>− Weight increase<br>− Inapplicable for all waste types<br>− Risk of exposure<br>− Unknown air emissions |
| Chemical disinfection | − Volume reduction<br>− Time efficient<br>− Removal of waste odour | − High capital cost<br>− Inapplicable for all waste types<br>− Unknown air emissions<br>− Storage and handling of chemicals |

#### 3.4.1. Incineration

Incineration is the most widely practised treatment method due to its applicability to treating all waste types [23]. The incineration process is carried out in furnaces operated at

temperatures of 800–1200 °C [23]. The high temperatures kill the pathogens, destroy 90% of organics, and change the waste characteristics such as weight, volume, and shape [23]. This process is governed by several parameters such as [21]:

- Mixing of waste,
- Moisture content,
- Amount of waste in the furnace,
- Temperature,
- Residence time,
- Maintenance and repair.

Incineration produces fly ash and emissions such as dioxins, furans, and mercury [23]. Dioxins and furans are considered carcinogenic, have a half-life ranging from 7 to 11 years, and are persistent footprints on the environment [23]. Dioxin emissions can be reduced if the complete combustion of waste is achieved [23]. The dioxins emitted can also be treated using selective non-catalytic reduction (SNCR) [24]. This technology depends on the production of free nitrogen via the reaction between nitric oxide and ammonia, and this gas is considered to be of high effectiveness and low cost [24]. Mercury represents 3–9% of the emissions from incineration, which impacts the nervous system and general health [23]. Fly ash is the solid residue from incineration, rich in heavy metals [25]. Fly ash can be recycled but has to undergo chemical pre-treatment first by using ethylene diamine tetra acetic acid disodium (EDTA) or sodium sulphide, which removes the heavy metals from the fly ash [25]. Approximately 3 kg of $CO_2$ is produced from burning 1 kg of clinical waste, therefore, incinerating MW contributes to global warming by releasing significant amounts of greenhouse gases (GHG), mainly $CO_2$ [26].

### 3.4.2. Autoclave Disinfection

Autoclave disinfection is a treatment method using temperature and steam simultaneously to kill microbes [15]. It is operated at a lower temperature than incineration but with pressure and steam influence to achieve disinfection [20]. The operating conditions are 60 min, at 121 °C and 1 bar, followed by a cycle of 60 min at 134 °C to ensure the complete disinfection of waste [20]. The following aspects govern the operation of the autoclave [21]:

- Temperature (121–134 °C),
- Steam penetration,
- Waste load,
- Length of the treatment cycle,
- Chamber air removal.

Due to the low operating temperatures of autoclaving, the waste appearance does not change, and the pathogens are not removed, which requires pre-treatment of the waste by incineration to be disposed of in landfills [15]. Thus, the autoclave is not optimum for all waste types.

### 3.4.3. Microwave Disinfection

Microwave disinfection uses low temperature and high microwaves for the reverse polymerisation and degradation of organic substances and microorganisms [23]. The waves induce molecular bond vibrations, saving energy and preventing emissions, making it a more environmentally friendly method [23]. The disinfection is operated at temperatures ranging between 177 and 540 °C electromagnetic waves of wavelength ranging between 1 mm and 1 m and frequency ranges between 300 and 3000 MHz [23]. Microwave disinfection has high costs and can be combined with incineration and autoclave [23]. The following aspects govern the operation of this method [21]:

- Waste characteristics,
- Moisture content,
- Microwave source strength,
- Exposure time,

- Degree of waste mixing.

### 3.4.4. Chemical Disinfection

Chemical disinfection is used to kill microorganisms and fight off pathogens by using chemicals [20]. It is primarily used for treating liquid infectious wastes such as blood, urine, faeces, or hospital sewage [20]. The chemical disinfectants that are commonly used are bleach solution (1%) or a diluted active chlorine solution (0.5%) [20]. In addition, other disinfectants such as lime, ozone, ammonium salts, and peracetic acid can be used [20]. This treatment method directly affects those in charge of the treatment due to the inhalation of volatile chemicals or irritations to the skin and eyes [20]. The following aspects govern the effectiveness of this method [21]:

- pH,
- Contact time,
- Waste and chemical mixing,
- Recirculation versus flow.

The residues of this treatment are liquid and solid residues [23]. The liquid residues are disposed of in the sewer system, and solid residues are disposed of in the landfill [23]. Determining an adequate treatment method involves defining the waste characteristics, type, and the desired outcome, which should be checked against each treatment method to show the advantages and shortcomings of each process [21].

### 3.5. Waste Recycling

Waste recycling is using produced waste or by-products for the same or different purposes. Most of the waste produced by the medical sector is non-hazardous waste that is mainly disposed of in landfills. The use of waste dumps and landfills can be reduced by recycling used products such as plastics, batteries, paper, glass, metals, and silver used in photographic processing [20]. Food and organic waste can be used for composting purposes [20]. Fly ash from incineration can be used after treatment in concrete mixtures and as building blocks [25]. The heat produced from the incineration could be used to heat water for a centralised heating system [20].

### 3.6. Waste Disposal

Rejects of the previous steps are transported to a sanitary landfill for disposal. However, landfills are not the optimum solution for handling MW due to their environmental effects. These effects are soil and water pollution caused by leachate and gas emissions into the air due to waste degradation [23]. Thus, the waste being disposed of should be minimised to the most, and achieving a circular economy guarantees that. Long-term decomposition of waste is the primary process responsible for waste disposal in landfilling [23]. Preventive measures should be taken to ensure the safe disposal of MW, which are [20]:

- Rapid cover of waste,
- Burying it under the old municipal waste of minimum burial of three months,
- Waterproof bottom,
- Minimum 2 m above the water table,
- No disposal of chemicals.

## 4. Case Studies

### 4.1. Germany

The disposal of health care wastes has to be considered from the perspective of statutory and occupational safety, in addition to available workflows of the health care facility and economic demands. Many European Union (EU) countries, including Germany, have implemented waste management systems. In 2012, Germany identified 50 types of hospital waste, including electronic and electrical waste of around 4500 tons annually or

12 tons daily. Only 1–3% or 60,000–80,000 tons of total waste were infectious MW, with 5–10 times higher disposal costs than regular MW [27]. Another study stated that 75% to 95% of MW is comparable to domestic waste that is neither contaminated by body fluids nor by chemicals and has no sharp waste [28].

In 2013, the EU issued a regulation called "The European Eco-Management and Audit Scheme (EMAS)" to improve the environmental performance of business organisations. The businesses that met the EMAS requirements were awarded the EU label. At the beginning of 2012, a total of 1007 organisations in Germany were registered/certified under EMAS. The principal mandate of most of these companies is to attain better waste management and resource efficiency performance. Approximately 7% of these organisations belong to the health sector [27].

Policies

The waste management system in Germany is well-developed, following numerous laws and principles. Below are some guiding acts and regulations [15,28]:

- "The Directory on Hazardous Waste (94/904/EC), 1994",
- "European Waste Catalogue (EWC)" and "European Commission Decision 2000/532/EC-2000",
- "Closed Subsistence Cycle Waste Management Act"—Principle law of waste management,
- "Infection Control and Safety Regulations",
- "The Dangerous Goods Regulations"—for the transport of hazardous waste,
- "Federal Control of Pollution Act"—for the incineration of waste.

Germany is the leader in waste recycling in Europe and uses a unique waste categorisation system. About 75% of all incineration facilities in Europe exist in The Netherlands, UK, Germany, France, Italy, and Sweden. During COVID-19, Germany recommended segregation and collection of all MW, except sharps or glass, in a container, as raw materials of non-recyclable nature. Numerous solutions, such as incinerators using modern technologies (Germany, Sweden, Japan) and modern landfills (Denmark, Norway, Sweden), are used for the disposal of healthcare waste [29].

In Germany, hospitals and other health care facilities are obliged to nominate one responsible official for the proper disposal of waste. The person is accountable for following occupational safety and legal requirements. In addition, the segregation of waste from the source and their proper disposal is strictly maintained under facility quality management. Hospitals must be proofed of the correct disposal of hazardous waste by their authorities. Depending on the MW category, the collection and disposal of waste are considered independently. Some MW disposal practices are described below [28]:

- Sharp waste: is immediately collected in resistant leak-proof and closable containers at the point of the waste generation source. This type of MW is disposed of together with general waste;
- Anatomical waste: is also collected in resistant leak-proof and closable containers from the point of waste generation origin and refrigerated and transported for incineration by authorised plants;
- Infectious waste: is also collected from the source of origin in protected containers, flagged with a biohazard symbol. Authorised companies transport waste to approved standard incineration plants;
- Other waste: waste produced during care or treatment without considering specific requirements for disposal is also collected and finally disposed of with domestic waste following municipal regulations;
- Amalgam waste: this is waste collected from dentistry and is collected separately and recycled by the producer or distributor;
- Chemical waste: is collected in a leak-proof container, flagged by hazardous properties of chemical content, which a specialised waste management company manages;
- Cytostatic drug wastes: are also collected at the point of waste generation origin in leak-proof containers flagged with a danger symbol. These wastes are transported to the authorised incineration facilities by companies specialising in MW transportation.

## 4.2. China

Like many developing countries, China had not paid enough attention to managing MW within the country before 2003. At the end of or after 2003, the government of China issued many acts and regulations to address MW adequately, including collection, transportation, and temporary storage by the MW's generator [30]. Nanjing, for example, has rapidly developed cities, like other cities in China in terms of public and private medical care establishments. Nanjing's hospital beds increased from 17,000 to around 20,000 between 1997 and 2006, while the number of health care facilities also increased from 1301 to 2085 during the same period. At that time, 159 hospitals existed in this city. Generation of MW ranges from 0.5 to 0.8 kg/bed day with a weighted average of 0.68 kg/bed day. Around 73% of the hospitals segregated and collected their MW at the point source of origin. Finally, a centralised disposal system has been established, and the disposal cost of MW was around USD 580/ton [31].

Policies

China follows the below-mentioned policy framework for MW management [30,31]:

- "Medical waste control act 380"—Mandatory obligation and requirements of a disposal facility for the MW generator;
- "Regulation 287"—related to the MW category issued by the Ministry of Health and State Environment Protection Administration;
- "Administrative Penalty Regulation 21"—deals with the behaviour and the improper management of MW by the generator;
- "Standard HJ 421-2008"—concerned about the standards regarding containers, packaging bags, and warning labels and symbols of different types of MW;
- "Technical Standard for Medical Waste Incinerator, 2003";
- "Technical Specifications for Centralised Incineration Facility for Hazardous Waste, 2005";
- "Measures on Permit for Operation of Hazardous Wastes, 2004";
- "Regulations on the Administration of Medical Wastes, 2003";
- "Standard for Pollution Control on Hazardous Waste Storage, 2001";
- "Pollution Control Standard for Hazardous Wastes Incineration, 2001";
- "Standard for Pollution Control on the Security Landfill Site for Hazardous Wastes, 2001";
- "Measure for the Administration of Registration of Hazardous Chemical, 2002".

The year 2005 witnessed the generation of 740,000 tons of MW in China. Only 10% of that waste is well managed., while 90% of the MW was disposed of in municipal sewage systems or discharged without control. Mostly, MW was incinerated with primitive technology and simple equipment. In 2004, the government of China invested 1.9 billion USD in deploying seven monitoring centres for dioxin and one MW disposal centre in each central city. In addition, 13 comprehensive treatment and disposal centres were planned to be constructed in each province under this initiative. By the end of 2004, 177 formal disposal centres were established for industrial hazardous waste management. Most provinces have established an industrial hazardous waste disposal centre for sound waste management [31].

## 4.3. United States of America

Around 20–30% of the healthcare sector waste in the US is produced from surgeries in operating rooms, which accounts for more than 1.8 billion kg of waste. Most waste comprises disposable (single-use) materials, equipment, and sterile packaging. For example, the waste produced from one routine operation is more than that produced by a family of four in a whole week [5].

The disposal of MW was considered under state regulation before 1988. However, public concern started to build up during the summer of 1987 and 1988 due to MW wash-up onto the beaches along the east coast from Maine to Florida, the west coast, the Great Lakes, and the Gulf Coast. The before-mentioned and other incidents made the federal government pay more attention to the environment and improve MW management [32,33].

As the world's top medical waste-producing country, the USA generates over 3.5 million tons of MW annually with an average disposal cost of 790 USD/ton. In addition, the USA's health care facilities generated approximately 10.7 kg/bed/day of waste, including infectious waste of 2.79 kg/bed/day [15,32]. Furthermore, with the prevalence of COVID-19, the production of KN95 masks increased four times compared to the output before the pandemic, adding more waste and causing further pollution [34].

Policies

The treatment of MW is under the control of governmental regulatory bodies, with strict public laws and regulations. The USA follows the regulatory framework described in Table 3 for managing and disposing of MW based on the waste category. The main aim of the MW Tracking Act (MWTA) was to trace and control the regulated hospital waste from the source of origin to the point of disposal and to set separation requirements, handling, and labelling of the hospital wastes by implementing a two-year demonstration programme (22 June 1989–22 June 1991) in five states (Connecticut, New Jersey, New York, Rhode Island, and Puerto Rico). Under the Act, the required responsibility was shared between the Environmental Protection Agency (EPA) and the Agency for Toxic Substances and Disease Registry (ATSDR). The EPA's primary responsibility was promulgating regulations for segregation, packaging, labelling, and tracking the MW. The ATSDR was responsible for reporting hospital waste's effects on health to Congress [32].

**Table 3.** Regulatory framework and categorisation of medical waste in the USA.

| No. | Type of Waste Category | Name of the Regulatory Framework |
|---|---|---|
| 1. | Regulated medical waste | Medical Waste Tracking Act (MWTA), 1988; States Regulations and EPA Guidelines; Resource Conservation and Recovery Act (RCRA) (40 Code of Federal Regulations (CFR) 240.101) |
| 2. | Non-regulated medical waste | States Regulations |
| 3. | Hazardous waste | RCRA (40 CFR 260–265 and 122–124) and States Regulations |
| 4. | Radioactive waste | Nuclear Regulatory Commission (NRC) Standards (10 CFR 20) |

The USA has invented a new patented technology by the team of engineers in the Idaho National Laboratory, Idaho Falls, ID, USA. It helps to achieve better treatment and management of MW. Following this, Med-shred, Inc. (Houston, TX, USA) has developed a mobile shredding and chemical disinfectant device to help reduce the amount and onsite treatment of hazardous MW. The MW can be shredded by the machine and converted into disposable municipal waste, which is then wetted with the disinfectant spray and immersed in a disinfectant solution. After that, the waste is to be transferred to a drying chamber for drying using hot off-gas [35,36]. Another research study showed that four types of disposal mechanisms have been followed for managing MW in the USA [33]. These are;

- Incineration: three types of incineration are used for MW disposal. These are controlled air, multiple chamber air, and rotary kiln models. The Environmental Protection Agency (EPA) estimates that more than 70% of the total MW generated in the USA is incinerated [32];
- Steam sterilisation or autoclaving: it is necessary to follow sterilisation or autoclaving before landfill disposal of MW. Autoclaving includes keeping the temperature between 120 and 135 °C, bags of infectious MW are placed in a chamber and steamed for 30–50 min. After that, the sterile waste can be safely disposed of in a landfill [37].

Another study showed that 49–60% of healthcare waste is incinerated, 20–37% is autoclaved, and 4–5% is disposed of using other technologies. In addition, incineration standards for MW became stricter after the USA's amendments to the Clean Air Act (CAA) in November 1990. The amendments have established the emission guidelines and limits for, among other pollutants, mercury, dioxins, and furans [15].

*4.4. Egypt*

Egypt endeavours to promote its MW management facility within the country. However, the authorities have failed to establish an efficient system regarding waste segregation, collection, transfer, or treatment because of a lack of strong legislative enforcement. Furthermore, many governorates faced numerous difficulties in implementing integrated MW management. The MW generation from the hospitals in El-Beheira Governorate, Egypt was about 1249 kg/day, ranging from 11 to 52% of hospital waste. The rate MW generation ranges between 0.23 and 2.07 kg/bed/day with a mean of 0.85 kg/bed/day. The amount of MW generation rate from the public hospital (0.23 kg/bed/day) was lower than the private hospital (2.07 kg/bed/day) in this governorate [38].

Medical waste has been regulated inadequately in developing countries, especially by recyclers of informal sectors, due to the lack of coordination between the national and international strategies for the proper, safe, and sustainable recovery, treatment, and disposal of personal protective equipment (PPE) during COVID-19. For example, PPE plastic waste was found in Alexandria and Hurghada, two coastal cities of Egypt. Gloves and face masks accounted for 38% and 57%, while plastic bags were 18.3% and 7.0% of total marine litter collected in the two cities, respectively [39]. Another study shows solid waste generation from health care facilities in Egypt increased significantly from 70 to 300 tons/day during COVID-19 [40].

Policies

There are no solid regulations for the sound management of MW in Egypt. However, according to the recommendation from WHO (1999) and the Egyptian Prime Minister (1994) Executive decree no 338/1995 and no 1741/2005 of Environmental Law No 4, Egypt tried to implement systematised integrated hospital waste management [38].

In Egypt, the management of MW in any of the studied healthcare facilities was not under strict regulations due to the lack of definite policies and standards. Safety engineers, senior nurses, and workers or department aid workers are usually responsible for collecting, transporting, and managing biomedical waste. They use a trolley or cart, which is not specially designed to handle and move the waste to the storage area of the hospital. No protective gear/measures were followed by the staff engaged in MW handling. Only the sharp waste is collected in puncture-resistance containers, while other medical wastes are packed in plastic bags with or without segregation. No specific symbol or colour is used to determine the healthcare waste packaging. Of the surgical, medical, and laboratory departments, 60% store their biomedical waste inside the utility rooms; the remaining 40% is stored in intensive care units. In addition, a general storage area is located on the basement floor near the exit or incinerators in all hospitals. Generally, the labour, operating rooms, and daily units are immediately transported to the shared storage area, and waste is disposed of daily. The average storage period was between 4 and 8 days [41].

The legislations from the Ministry of Environmental Affairs administer MW disposal along with the Ministry of Health and Population. The regulations classified the waste generated from the health care facilities. Some of these were considered as hazardous waste and needed to be cautiously handled during the source segregation, collection, transportation, handling, and final disposal [41]. The waste generated from hospitals is treated by incineration or autoclaving after segregation. A portion of the non-reusable waste is used for fertiliser production, and the remaining waste is transported to dumping places with pre-designed landfill cells for final disposal. Methane gas is collected from the waste using wells and pipes channelled into the networks [42]. Lack of environment-friendly collection and management, including littering, disposal in uncontrolled landfills, and open dumps of COVID-19 wastes, may worsen the current plastic pollution in Africa (15 out of 57 countries, including Egypt) [43]. Another study mentioned that MW was generated from hospitals in Egypt at an average of 70.5 tons/day. Less than 35 tons were incinerated safely and efficiently, and the rest of the waste was either traded illegally for recycling or disposed of with municipal solid waste [40].

*4.5. Management of MW in Other Countries*

4.5.1. India

The Indian Ministry of Environment and Forests issued the first set of guidelines to monitor and control MW in 1998; these laws underwent a number of modifications in subsequent years. In 2016, the Ministry of Environment, Forestry, and Climate Change amended the regulations in an effort to reduce environmental contamination. The coverage was expanded, categorization and authorization were simplified, techniques for segregation, transportation, and disposal were enhanced, and emission standards were stricter (reducing the acceptable concentration of suspended particulate matter emitted from incinerators to 50 mg/nm$^3$) [44]. An estimated 0.33 million tons of MW were generated annually in India as a result of approx. 0.5–2.0 kg/bed/day of waste from healthcare facilities [45].

4.5.2. Canada

Medical waste disposal is undeniably not covered by a national regulatory framework, owing, in part, to the previously indicated Canadian jurisdictional differences on health-related matters [46]. The majority of provinces base their management of MW on standard regulations that apply to all forms of solid waste. Only Quebec appears to have laws that specifically address biological waste, such as the "Environmental Quality Act" and the "Act respecting certain measures enabling the enforcement of environmental and dam safety legislation". Quebec had rules and regulations for managing and processing MW beginning with the use of materials and progressing to segregation, collection, transportation, and treatment [47].

4.5.3. Europe

Member states in the European Union (EU) are responsible for enacting legislation that complies with and serves to implement European Commission (EC) directives on waste related laws, directives, and standards. As a result, the EC has urged the EU member states to categorize medical wastes in line with Chapter 18 of the European Waste Catalogue (EWC), where it has prepared a list of waste descriptions for the various components MW. The EWC was established on its own in the year 2000 by the European Commission Decision 2000/532/EC [15].

4.5.4. Australia

There are no rules or regulations in Australia that are applicable on a national scale that deal with MW. At the level of the individual states and territories, each state and territory are responsible for developing its own policies and laws to manage municipal solid waste as well as other types of waste; nevertheless, these policies and regulations are required to be in accordance with international conventions/regulations [48]. For example, the PVC waste, which can include oxygen tubing, masks, and IV bags, contributes to Australia's annual production of 15 tons of MW. Some of the solid wastes containing PVC is processed locally, but the majority of it is shipped to other nations to undergo additional processing [49]. The related policies in Australia are as follows [48]:

- The "Clinical and Related Waste Management Policy of 2016" in Western Australia addresses both the hazardous and non-hazardous types of MW,
- The "Environment Protection (Waste Management) Policy 1994" addresses and regulates all types of waste in southern Australia.

## 5. COVID-19 and Medical Waste

Most countries worldwide recorded a significant decrease in air pollution and greenhouse gas emissions during the first months of the pandemic, mainly due to the limitations on or elimination of many human activities. Meanwhile, the generation of MW from discarded, used equipment and materials, including all sorts of plastic containers, test kits, gowns, and syringes, significantly increased after the COVID-19 outbreak and the making and distribution of vaccines [50].

In China alone, approximately a 24% increase in MW generation was detected, when it peaked at more than 6000 tons in 2020 during the COVID-19 pandemic. A second study found that during the COVID-19 outbreak in China, the amount of MW decreased by up to 30% in medium and large cities; however, the generation of MW containing high plastic content has increased by ~ 400% in Hubei province [51]. Furthermore, the city of Wuhan generated more than 240 tons/day of MW after the pandemic started, which is higher than the usual amount by 190 tons compared to the situation before the pandemic [52].

Waste resulting from medical practices and the use of personal protection equipment, especially after the COVID-19 pandemic, is considered a source of the spread of infection. For example, the rate of MW due to the Coronavirus disease in Wuhan, the pandemic's epicentre, has increased from 0.6 kg/bed/day to 2.5 kg/bed/day [53]. In Bangladesh, for instance, the level of plastic pollution increased drastically due to the increase in plastic waste generation after the wide spread of COVID-19, in which more than 14,000 tons of biological and MW compared to about 200 tons of waste per day before the virus [54]. A study reviewing the waste situation, including biomedical waste, after the COVID-19 outbreak, found that besides the increase in MW generated from healthcare facilities during the pandemic, all waste can be considered infectious MW because of the high numbers of COVID-19 cases recorded. Furthermore, with the wide spread of the virus worldwide at different periods alongside the high infectivity rate of the virus, the possibility of cross-contamination and the spread of the disease increased dramatically [55].

From a practical perspective, in order to find the most appropriate method of treating medical wastes, Voudrias [56] examined five distinct techniques, including chemical disinfection, steam disinfection, microwave disinfection, reverse polymerization, and incineration. The goal was to select the most effective approach through the use of multicriteria analysis. When evaluating the various technologies, the author suggested using the analytic hierarchy process, which prioritizes the consideration of environmental, economic, technological, and social factors as the primary evaluation criterion. The author recommended the following sub-criteria for the environmental part: emission of greenhouse gases, environmental impact of air emissions, solid wastes and liquid residues, energy and water consumption pattern, volume reduction, and inactivation of microorganisms. The author suggested considering the capital, operational, maintenance, and disposal costs for the economic criteria, while for the technical criteria, the treatment effectiveness, automation, and the necessity for qualified operators was recommended. Moreover, it was also acknowledged that the acceptance of both the technology and the cost should serve as the sub-factor for the social criteria. In order to deploy these technologies in a more realistic setting, the author suggested considering a number of additional site-specific variables, such as the maximum load that can be treated by the selected technology and the local regulatory requirements.

## 6. Conclusions and Recommendations

Medical waste contributes to a considerable percentage of the total waste generated in most countries, and about 75% of MW is non-hazardous. The rest is considered to be hazardous since it is contaminated with infectious contaminants that can cause illness and transmit various diseases; therefore, proper handling and treatment of MW are needed. Better management can be implemented with appropriate (local) laws and regulations to reduce the risk of cross-contamination and decrease levels of emitted pollution from treatment and recycling of this type of waste using incineration, considered the most widely applied method for treating MW in the world. In addition, the COVID-19 outbreak resulted in a massive surge in MW, especially personal protection equipment (PPE), e.g., masks, gowns, and vaccination needles and syringes that need great attention due to their dangerous environmental impacts and low degradability.

In order to enhance MW management and treatment and make it more efficient and less damaging to the environment and to reduce the cost of production, disposal, and treatment, several aspects have to be considered: (i) to reduce the quantity of waste

generated by regulating the use of materials and disposable equipment, (ii) to segregate the waste according to the regulations, with more strictness and attention, (iii) to limit the use of incineration, (iv) to follow stricter technological measures for the incineration of MW, e.g., filtration and treatment of emissions from the incinerators, (v) to invest in new eco-friendly technologies for the disinfection and treatment of MW, and (vi) to develop more readily degradable materials for the purpose of producing personal protection equipment.

**Author Contributions:** Conceptualization, M.A., A.E., D.A. and E.R.R.; methodology, M.A., A.E. and D.A.; formal analysis, M.A., A.E., D.A. and E.R.R.; investigation, M.A., A.E. and D.A.; writing—original draft preparation, M.A., A.E., D.A. and E.R.R.; writing—review and editing, M.A., A.E., D.A. and E.R.R.; supervision, M.A. All authors have read and agreed to the published version of the manuscript.

**Funding:** This research received no external funding.

**Data Availability Statement:** Not applicable.

**Acknowledgments:** Elmanadely, A. acknowledges the Dutch government for providing the MENA (The Middle East and North Africa) fellowship (EB/MRA/FINAD/1090264) to pursue her MSc degree (2021–2023) at IHE Delft, The Netherlands. Akter, D. acknowledges the Dutch government for providing the Orange Knowledge Programme (OKP) fellowship (Ref: EB/MRA/FINAD/1087171) to pursue her MSc degree in Environmental Science with a specialization in Environmental Science and Technology (2021–2023) at IHE Delft, The Netherlands.

**Conflicts of Interest:** The authors declare no conflict of interest.

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
