# Peer review of "A Review on Medical Waste Management: Treatment, Recycling, and Disposal Options"

_environments, doi:10.3390/environments9110146_

Round 1

Reviewer 1 Report

The paper discussed the classification of the various types and categories of medical waste and its treatment methods are discussed. Due to the fact that medical waste is contaminated and hazardous, it must be managed and processed using complex steps and procedures. The primary medical waste treatment method is incineration, which is regarded as a highly polluting process that emits numerous pollutants that degrade air quality and pose a threat to human health and the environment. As case studies, medical waste treatment and disposal practices in Germany, China, USA, and Egypt were compared, and the legislations and laws enacted to regulate medical waste in each of these countries are reviewed and discussed.

The paper presents the basic data on medical waste management in a comprehensive and exhaustive manner in the world. Examples of their management in several countries and relevant legal regulations binding on them were also discussed. Data on the incineration of medical waste are particularly interesting.

This is a very good article and I recommend publishing it.

Author Response

Authors response: The authors thank the reviewer for the very nice description of the paper, good comments, and recommendation. 

The authors have revised the manuscript and incorporated all the comments of the reviewer. The revised text has been highlighted in yellow color and important recent references pertaining to medical waste have also been added to the revised manuscript.

Reviewer 2 Report

The article is a significant contribution to the medical waste management problem. However, as a review article, it is not expected that test results will be addressed to contribute to using these residues.

It would be interesting in future articles if this approach were used. Here's a suggestion.

Author Response

(The authors gave the same response as above.)

Reviewer 3 Report

This paper introduces the classification of medical waste, the treatment process and the current status of medical waste treatment in different countries and gives recommendations for the management and treatment of medical waste.This topic and the corresponding learning area are very meaningful and worthy of further study.

Here are some minor issues that may be considered for improving this paper:

1.section 3.4 mentions several common methods of treatment of medical waste and their advantages and disadvantages. Is it possible to give a more comprehensive framework for selecting the appropriate method for medical waste?

2.section 4. only describes the current status of medical waste treatment and related policies in four countries, however, there is no specific comparison of the status between different countries. There is a little confusion about what these cases are meant to express?

Author Response

(The authors gave the same response as above.)

Reviewer 4 Report

In the article presented for review, classification of the various types and categories of medical waste and its treatment methods are discussed.s case studies, medical waste treatment and disposal practices in Germany, China, USA, and Egypt were compared, and the legislations and laws enacted to regulate medical waste in each of these countries are reviewed and discussed.

I read the article submitted for review with great interest. I have a few comments:

1. In the title of the article, please write the word "treatment" with a lowercase letter.

2. Is it known what is the share of plastics from medical waste in of the total global carbon budget which contributes to contributed to the greenhouse gas (GHG) emissions resulting from the life cycle of plastics? Please complete it, if possible, on lines 37-50.

3. It is a pity that the case studies only cover four countries. I encourage the authors to analyze other countries.

Therefore, the publication is a valuable source of information to implement new solutions in this field and forms the basis for further research i

Thank you for considering my opinion. I encourage the authors to continue working on improving the manuscript.

Author Response

(The authors gave the same response as above.)
